

# How open is the asteroid-mass primordial black hole window?

Matthew Gorton[⋆] and Anne M. Green[†]

School of Physics and Astronomy, University of Nottingham,
Nottingham, NG7 2RD, United Kingdom

⋆ matthew.gorton@nottingham.ac.uk , † nne.green@nottingham.ac.uk

## Abstract

Primordial black holes (PBHs) can make up all of the dark matter (DM) if their mass, $m$, is in the so-called 'asteroid-mass window', $10^{17}\,\mathrm{g} \lesssim m \lesssim 10^{22}\,\mathrm{g}$. Observational constraints on the abundance of PBHs are usually calculated assuming they all have the same mass, however this is unlikely to be a good approximation. PBHs formed from the collapse of large density perturbations during radiation domination are expected to have an extended mass function (MF), due to the effects of critical collapse. The PBH MF is often assumed to be lognormal, however it has recently been shown that other functions are a better fit to numerically calculated MFs. We recalculate both current and potential future constraints for these improved fitting functions. We find that for current constraints the asteroid-mass window narrows, but remains open (i.e. all of the DM can be in the form of PBHs) unless the PBH MF is wider than expected. Future evaporation and microlensing constraints may together exclude all of the DM being in PBHs, depending on the width of the PBH MF and also the shape of its low and high mass tails.

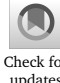

# 1   Introduction

Primordial black holes (PBHs) may have formed in the early Universe [1–3] and are a potential dark matter (DM) candidate. The most commonly-studied PBH formation mechanism is the collapse of large density fluctuations generated by inflation. The abundance of PBHs is constrained by observations over a wide range of PBH masses (see Refs. [4–7] for recent reviews). Under standard assumptions, PBHs can only account for all the DM if their mass $m$ lies in the range $10^{17}\,\mathrm{g} \lesssim m \lesssim 10^{22}\,\mathrm{g}$, often referred to as the 'asteroid-mass window'. Lighter PBHs making up all of the DM is excluded by limits on the products of their evaporation, while planetary and Solar mass PBHs are excluded by microlensing and other observations.

Typically observational constraints are calculated assuming that PBHs have a delta-function mass function (MF). However, PBHs formed from the collapse of large density perturbations are expected to have an extended mass function. Due to critical collapse, the mass of a PBH depends on both the horizon mass and the amplitude of the density fluctuation from which it forms. Consequently even if PBHs all form at the same time, from a narrow peak in the primordial power spectrum, they have a range of masses [8–10]. Furthermore, the peaks in the primordial power spectrum that are produced by concrete inflation models, for instance hybrid inflation with a mild waterfall transition [11], can be broad.

With an extended PBH MF the constraints are 'smeared out'; for each constraint the tightest limit on the fraction of DM in the form of PBHs, $f_{\mathrm{PBH}}$, is weakened, however the range of peak masses[1] for which $f_{\mathrm{PBH}} = 1$ is excluded is wider [12–14]. Refs. [12–14] calculated constraints on PBHs with a lognormal MF, which provides a reasonable fit to the MFs found for PBHs produced by various inflation models [12, 15]. However, Gow et al. [16] found that the MFs they calculate are better fit by other functions, specifically a skew-lognormal distribution and a form motivated by critical collapse. As Gow et al. [16] mention, the shape of the low mass tail is important when considering evaporation constraints on PBHs with MFs which peak in the asteroid-mass window. It is important to ascertain how the extent to which the asteroid-mass window remains open (i.e. for what range of peak masses $f_{\mathrm{PBH}} = 1$ is allowed) depends on the shape of the PBH MF.

We recalculate constraints on $f_{\mathrm{PBH}}$ in the asteroid-mass window for the MF fitting functions presented in Ref. [16]. Sec. 2 presents the constraints we consider, the fits to the PBH MF that we use and their time evolution, and the method for applying the constraints to extended mass functions. We present the current and prospective future constraints on PBHs with extended MFs in Sec. 3 and conclude with discussion in Sec. 4.

# 2   Method

In Sec. 2.1 we overview the (current and proposed future) evaporation and stellar microlensing constraints that we use. In Sec. 2.2 we overview the best-performing MF fitting functions from Ref. [16] and the evolution of the MF due to evaporation, and in Sec. 2.3 we outline how the constraints are applied to extended mass functions.

## 2.1   Constraints on PBHs around the asteroid-mass window

The constraints on PBHs with masses $m \lesssim 10^{17}\,\mathrm{g}$ and $m \gtrsim 10^{22}\,\mathrm{g}$ arise from PBH evaporation via Hawking radiation (Sec. 2.1.1) and stellar microlensing (Sec. 2.1.2) respectively. The constraints that we consider (both existing and prospective) are shown in Fig. 1 for a delta-function PBH MF.

---

[1]Here and throughout peak mass refers to the mass at which the mass function is maximal.

### 2.1.1 PBH evaporation

PBHs formed from the collapse of large density perturbations rapidly lose angular momentum and charge [17, 18]. Hawking [19, 20] showed that a non-rotating, uncharged black hole (BH) of mass $m$ radiates with a temperature

$$k_B T_{BH} = \frac{\hbar c^3}{8\pi G m} = 1.06 \left(\frac{m}{10^{16}\,g}\right)^{-1} \text{MeV}, \tag{1}$$

where $c$ is the speed of light, $G$ is the gravitational constant, $\hbar$ is the reduced Planck constant and $k_B$ is the Boltzmann constant.

As a result of Hawking radiation, in the absence of accretion or mergers, the mass of a BH decreases at a rate [18, 20]

$$\frac{dm}{dt} = -\frac{\hbar c^4}{G^2} \frac{\alpha(m)}{m^2}, \tag{2}$$

where $\alpha(m)$ parameterizes the number of particle species which can be emitted at a significant rate from a BH of mass $m$. BHs with mass $m \gg 10^{17}\,g$ can only emit photons and neutrinos, while those with smaller masses (and higher temperatures) can emit other particle species, such as electrons and positrons [18, 21]. Unstable particles emitted by PBHs decay to stable secondary particles, such as photons and electrons. The total emission of a given species from a PBH is the combination of the primary (i.e. directly emitted) and secondary components. As a result of Hawking evaporation, the PBH mass changes with time and PBHs have a finite lifetime. Since the PBH mass is time-dependent, the PBH mass function (MF) also evolves with time (see Sec. 2.2.2).

Constraints on the fraction of dark matter consisting of PBHs, $f_{PBH}$, can be obtained by comparing the flux of Hawking-emitted particles from PBHs with observations (see e.g. Ref. [28] and for a recent review Ref. [29]). For a delta-function MF, existing constraints exclude $f_{PBH} = 1$ for PBHs with mass $m \lesssim 10^{17}\,g$. There are various evaporation constraints, from different particle species and observations, calculated using different assumptions, with different uncertainties (see Ref. [29]). We consider two illustrative constraints: the INTEGRAL/SPI MeV gamma-ray limits from Ref. [23] which tightly constrain $f_{PBH}$ (for a delta-function MF) for $10^{16}\,g \lesssim m \lesssim 10^{17}\,g$, and the Voyager 1 $e^{\pm}$ limits from Ref. [22] which tightly constrain $f_{PBH}$ for $m \lesssim 10^{16}\,g$. As we will see in Sec. 3, the constraint that rules out $f_{PBH} = 1$ for the largest value of $m$ doesn't necessarily exclude $f_{PBH} = 1$ for the largest peak mass for broad, extended MFs.

Ref. [23] calculates constraints from gamma rays produced by positrons from PBH evaporation annihilating with electrons in the interstellar medium, including the contribution from positrons that first form a positronium bound state. We use the constraint (shown with a dashed purple line in Fig. 1 of Ref. [23]) obtained using the INTEGRAL/SPI limit on the flux of MeV gamma rays from a component with a Navarro-Frenk-White (NFW) density profile from Ref. [30]. We use the Voyager 1 $e^{\pm}$ constraint in the left panel of Fig. 2 of Ref. [22] with astrophysical background subtraction for their propagation model A, which has strong diffusive reacceleration.

Proposed future MeV gamma-ray telescopes have the potential to place tighter constraints on evaporating PBHs, extending the range of masses where $f_{PBH} = 1$ is excluded to larger $m$. As an illustrative case we consider the projected constraints from observations towards the Galactic Centre (assuming a NFW profile) by the proposed GECCO telescope from Fig. 9 of Ref. [26].[2]

---

[2]The proposed AMEGO telescope would be able to exclude $f_{PBH} = 1$ to somewhat larger $m$ [31, 32].

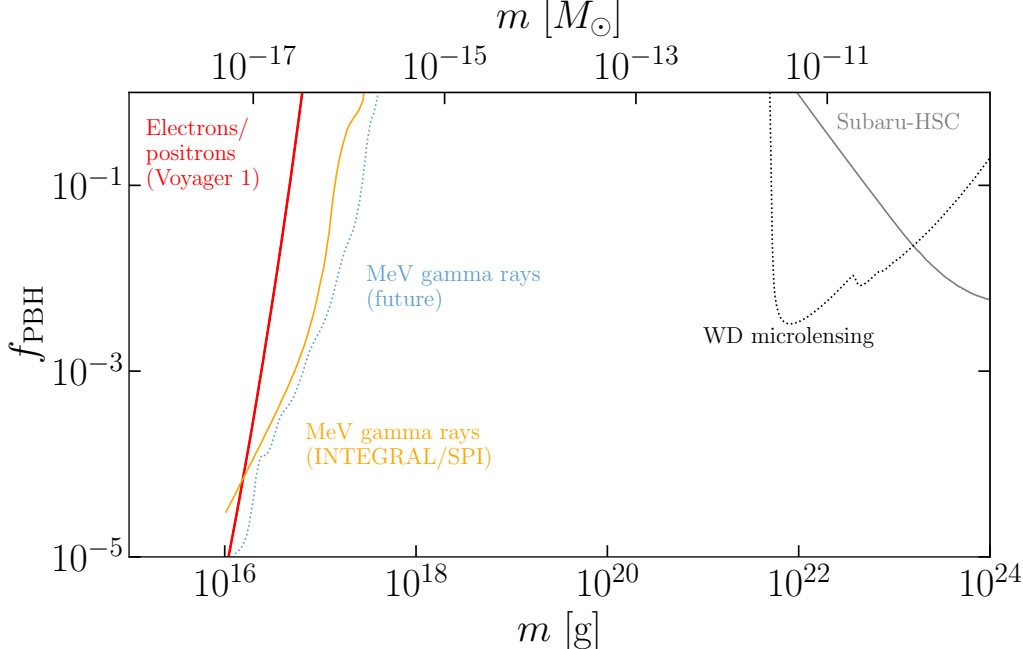

Figure 1: The constraints we use on the fraction of dark matter in PBHs, $f_{\text{PBH}}$, as a function of the PBH mass, $m$, assuming a delta-function PBH MF. Current constraints are shown as solid lines and prospective future constraints as dotted lines. The current evaporation constraints are from Voyager 1 measurements of the local flux of electrons and positrons [22] (red) and INTEGRAL/SPI observations of MeV gamma rays [23] (orange). The current stellar microlensing constraints in this mass range are from Subaru-HSC [24], as calculated in Ref. [25] (grey). The prospective future evaporation constraints are from a MeV gamma-ray telescope [26] (light blue) while the microlensing constraints are for a LMC white dwarf survey [27] (black).

### 2.1.2 Microlensing

Stellar microlensing is the temporary amplification of a background star that occurs when a compact object passes close to the line of sight to the star [33]. Observations of stars in M31 by Subaru-HSC [24] have been used to constrain $f_{\text{PBH}}$ in the mass range $10^{22}\,\text{g} \lesssim m \lesssim 10^{28}\,\text{g}$ [24, 25].

Accounting for the finite size of the source stars weakens the constraints significantly from those calculated assuming a point-like source [34, 35]. Additionally, for PBHs with masses $m \lesssim 10^{-11} M_\odot \sim 10^{22}\,\text{g}$, the Schwarzschild radius of the PBHs is comparable to the wavelength of the light used to observe the stars, resulting in diffraction and interference effects. Due to these 'wave optics' effects, it is not possible to constrain PBHs with $m \lesssim 10^{-11} M_\odot \sim 10^{22}\,\text{g}$ using microlensing surveys of main-sequence stars [27, 35]. We use the point-like lens constraint from Fig. 4 of Ref. [25].

To minimise the finite source and wave optics effects, Ref. [27] suggests a survey of white dwarfs in the Large Magellanic Cloud (LMC) using shorter wavelength observations. They find such a survey could place significantly tighter constraints on PBHs with mass $m \sim 10^{(22-23)}\,\text{g}$. We use the constraint from Fig. 8 of Ref. [27] that accounts for both finite source and wave optics effects.

## 2.2 PBH mass functions and evolution

### 2.2.1 Initial mass functions

The initial PBH mass function (MF) can be defined as

$$\psi(m_{\mathrm{i}}, t_{\mathrm{i}}) \equiv \frac{1}{\rho_{\mathrm{i}}} \frac{\mathrm{d}\rho(m_{\mathrm{i}}, t_{\mathrm{i}})}{\mathrm{d}m_{\mathrm{i}}}, \tag{3}$$

where $\rho(m_{\mathrm{i}}, t_{\mathrm{i}})$ is the comoving mass density in PBHs of initial mass $m_{\mathrm{i}}$ at the time they form, $t_{\mathrm{i}}$, and $\rho_{\mathrm{i}}$ is the initial total comoving mass density of PBHs. Due to near critical gravitational collapse [36], the mass of a PBH formed depends on the amplitude, $\delta$, of the perturbation from which it forms as well as the horizon mass, $M_{\mathrm{H}}$: $m_{\mathrm{i}} = k M_{\mathrm{H}}(\delta - \delta_{\mathrm{c}})^{\gamma}$, where $\delta_{\mathrm{c}}$ is the threshold for PBH formation, and $k$ and $\gamma \simeq 0.36$ are constants [8].[3] Consequently even if all PBHs form at the same time, i.e. from a delta-function peak in the primordial power spectrum, they will have a range of masses [8–10]. In this case the critical collapse (CC) initial MF is well approximated, assuming the probability distribution of the amplitude of the fluctuations is gaussian, by

$$\psi_{\mathrm{CC}}(m_{\mathrm{i}}, t_{\mathrm{i}}) = \frac{1}{\gamma m_{\mathrm{p}} \Gamma(\gamma + 1)} \left(\frac{m_{\mathrm{i}}}{m_{\mathrm{p}}}\right)^{1/\gamma} \exp\left[-\left(\frac{m_{\mathrm{i}}}{m_{\mathrm{p}}}\right)^{1/\gamma}\right], \tag{4}$$

where $m_{\mathrm{p}}$ is the mass at which the MF peaks and $\Gamma$ is the gamma function.

In reality the primordial power spectrum will have finite width, and PBHs will form on a range of scales. For various inflation models the MFs calculated, taking critical collapse into account, can be roughly approximated by a lognormal (LN) distribution [12, 15]:

$$\psi_{\mathrm{LN}}(m_{\mathrm{i}}, t_{\mathrm{i}}) = \frac{1}{\sqrt{2\pi}\sigma m_{\mathrm{i}}} \exp\left(-\frac{\ln^2(m_{\mathrm{i}}/m_{\mathrm{c}})}{2\sigma^2}\right), \tag{5}$$

where $\sigma$ is the width and $m_{\mathrm{c}}$ is the mean of $m_{\mathrm{i}} \psi_{\mathrm{LN}}(m_{\mathrm{i}}, t_{\mathrm{i}})$. The lognormal MF has been widely adopted as the canonical extended PBH MF (for instance when applying observational constraints to extended mass functions [13, 14]).

Gow et al. [16] investigated more accurate fitting functions for the initial MF of PBHs formed from a symmetric peak in the primordial power spectrum. They parameterise the peak in the power spectrum of the curvature perturbation, $\mathcal{P}_\zeta(k)$, as lognormal,

$$\mathcal{P}_\zeta(k) = A \frac{1}{\sqrt{2\pi}\Delta} \exp\left(-\frac{\ln^2(k/k_{\mathrm{p}})}{2\Delta^2}\right), \tag{6}$$

where $A$ and $\Delta$ are the amplitude and width of the peak and $k_{\mathrm{p}}$ is the comoving wavenumber at which it occurs. We have found that the broad peak in the primordial power spectrum produced by hybrid inflation with a mild waterfall transition [11] is fairly well-approximated by a lognormal with $\Delta \sim 5$. Ref. [16] calculates the PBH MF numerically as in Ref. [38], using the traditional (BBKS) peaks theory method [39] with a modified gaussian window function.[4]

---

[3]The criterion for PBH formation is traditionally specified in terms of the density contrast $\delta = (\rho - \bar\rho)/\bar\rho$. More recently it has been realised that the criterion is best specified in terms of the compaction function [37], and the dependence of the PBH mass on the compaction function has the same power law scaling.

[4]More recently Germani and Sheth [40] have formulated a procedure for calculating the abundance and MF of PBHs, using the statistics of the compaction function. They find (assuming a gaussian distribution for the perturbations) that the low mass tail of the MF is generically (i.e. for any primordial power spectrum) a power law, $\psi(m_{\mathrm{i}}, t_{\mathrm{i}}) \propto m_{\mathrm{i}}^{1/\gamma}$, while at large masses there is a cut off, which depends on the shape and amplitude of the power spectrum.

Gow et al. [16, 38] find that for narrow peaks in the power spectrum, $\Delta \lesssim 0.3$, critical collapse dominates the PBH MF; the MF is independent of the width of the power spectrum and skewed towards low masses. For $\Delta \gtrsim 0.5$ the width of the peak becomes important. As $\Delta$ is increased the width of the MF increases and the skew towards low masses decreases, and for large $\Delta$ (the transition occurs between $\Delta = 2$ and $5$) their MFs are skewed towards large masses [16]. Of the fitting functions considered in Ref. [16], the two that best reproduce this behaviour are the skew-lognormal and generalised critical collapse functions. The skew-lognormal (SLN) MF is a generalisation of the lognormal with non-zero skewness:

$$\psi_{\rm SLN}(m_{\rm i}, t_{\rm i}) = \frac{1}{\sqrt{2\pi}\sigma m_{\rm i}} \exp\left(-\frac{\ln^2(m_{\rm i}/m_{\rm c})}{2\sigma^2}\right)\left[1 + {\rm erf}\left(\alpha\frac{\ln(m_{\rm i}/m_{\rm c})}{\sqrt{2}\sigma}\right)\right], \qquad (7)$$

where $\alpha$ controls the skewness of the MF; for negative (postive) $\alpha$ the MF is skewed to low (high) masses.[5] The generalised critical collapse (GCC) MF[6] is given by

$$\psi_{\rm GCC}(m_{\rm i}, t_{\rm i}) = \frac{\beta}{m_{\rm p}}\left[\Gamma\left(\frac{\alpha+1}{\beta}\right)\right]^{-1}\left(\frac{\alpha}{\beta}\right)^{\frac{\alpha+1}{\beta}}\left(\frac{m_{\rm i}}{m_{\rm p}}\right)^{\alpha}\exp\left[-\frac{\alpha}{\beta}\left(\frac{m_{\rm i}}{m_{\rm p}}\right)^{\beta}\right], \qquad (8)$$

where $m_{\rm p}$ is the mass at which the generalised critical collapse MF peaks, and $\alpha$ and $\beta$ are parameters that control its behaviour in the low and high-mass tails respectively (for $m_{\rm i} \ll m_{\rm p}$, $\psi_{\rm GCC}(m_{\rm i}, t_{\rm i}) \propto m_{\rm i}^{\alpha}$). The generalised critical collapse MF is a generalisation of the critical collapse MF obtained assuming all PBHs form at the same time, Eq. (4), which corresponds to Eq. (8) with $\alpha = \beta = 1/\gamma$ [9]. Gow et al. [16] find that the generalised critical collapse MF is a better fit to their calculated MFs than the skew-lognormal for narrow peaks ($\Delta \lesssim 0.5$) where critical collapse dominates the PBH MF and it has negative skew, while for broad peaks ($\Delta \gtrsim 5$) the skew-lognormal is a better fit.

Ref. [16] focuses on stellar-mass PBHs. For the three fitting functions we consider (lognormal, skew-lognormal and generalised critical collapse), we choose values for the mass parameters ($m_{\rm c}$ or $m_{\rm p}$) in the asteroid-mass window. For the parameters which govern the shape of the MF ($\alpha$, $\beta$ and $\sigma$), we adopt the best-fit parameter values in Table II of Ref. [16][7] i.e. for simplicity we assume that these parameters do not depend on the PBH mass, or equivalently the position of the peak in the primordial power spectrum, $k_{\rm p}$. To facilitate comparison between the constraints obtained with different fitting functions, we present the constraints for the lognormal and skew-lognormal MF in terms of the peak mass, i.e. the mass at which the MF is maximal, $m_{\rm p}$. For the lognormal, $m_{\rm p} = m_{\rm c} \exp(-\sigma^2)$, while for the skew-lognormal there is no analytic expression for $m_{\rm p}$.

### 2.2.2 Time evolution of the mass function

The MFs presented in Sec. 2.2.1 are fits to the initial MFs calculated in Ref. [16]. For PBHs with initial mass $m_{\rm i} \lesssim 1 \times 10^{15}$ g, Hawking evaporation leads to significant ($> 10\%$) mass loss by the present day, and hence the MF varies with time [28, 41–43]. Therefore for extended MFs that are peaked at sufficiently small $m_{\rm p}$ and/or are sufficiently broad that there is a significant abundance of PBHs with initial masses $m_{\rm i} \lesssim 1 \times 10^{15}$ g, the time evolution of the MF should be taken into account.

---

[5]For consistency we use the same notation for the parameters of the fitting functions as Gow et al. [16], however the $\alpha$ parameters in the skew-lognormal and generalised critical collapse fitting functions affect their shapes in different ways.

[6]In Ref. [16], this is referred to as the 'CC3' model.

[7]Table II of Ref. [16] contains the best-fit parameter values for the skew-lognormal and generalised critical collapse MFs, we are grateful to Andrew Gow for providing those for the lognormal MF via email.

To evaluate the PBH mass today at time $t = t_0$, we follow Ref. [43] (see also Ref. [44]) and approximate $\alpha(m)$ as depending only on the initial mass, $\alpha(m) \approx \alpha_{\text{eff}}(m_i)$. Integrating Eq. (2), the PBH mass today, $m(t_0)$, can be expressed as

$$m(t_0) \approx \left( m_i^3 - 3 \frac{\hbar c^4}{G^2} \alpha_{\text{eff}}(m_i) t_0 \right)^{1/3},$$
(9)

where the formation time, $t_i$, has been set to zero since $t_0 \gg t_i$. Here, $\alpha_{\text{eff}}(m_i)$ is defined as

$$\alpha_{\text{eff}}(m_i) \equiv \frac{G^2}{\hbar c^4} \frac{m_i^3}{3 \tau_i},$$
(10)

where $\tau_i$ is the PBH lifetime (which can be calculated numerically e.g. using `BlackHawk` [45, 46]). This definition ensures that the PBH lifetime is calculated accurately for all initial masses $m_i$.

Using the conservation of the number of PBHs, the PBH MF today, $\psi(m, t_0)$, defined as

$$\psi(m, t_0) \equiv \frac{1}{\rho_i} \frac{d\rho(m, t_0)}{dm},$$
(11)

where $\rho(m, t_0)$ is the comoving mass density in PBHs with present day mass $m$, can be expressed in terms of the initial PBH MF, $\psi(m_i, t_i)$, defined in Eq. (3), as [28, 43, 47, 48]

$$\psi(m, t_0) = \left( \frac{m}{m_i} \right)^3 \psi(m_i, t_i).$$
(12)

The equivalent expression in Ref. [43] contains a factor that can be written as $(m/m_i)$ squared, rather than cubed, since they are considering number, rather than mass, densities.

## 2.3 Calculating constraints for extended MFs

We use the method introduced in Ref. [13] to apply constraints to extended MFs (a similar method is presented in Ref. [49]). The constraint on the fraction of dark matter in PBHs can be expressed as [13]

$$f_{\text{PBH}} \leq \left[ \int dm \frac{\psi_N(m, t_0)}{f_{\text{max}}(m)} \right]^{-1},$$
(13)

where $f_{\text{max}}(m)$ is the maximum fraction of dark matter in PBHs allowed for a delta-function MF, and $\psi_N(m, t_0)$ is defined as

$$\psi_N(m, t_0) \equiv \frac{1}{\rho(t_0)} \frac{d\rho(m, t_0)}{dm},$$
(14)

where $\rho(t_0)$ is the total comoving mass density in PBHs today. This definition of the MF is normalised to unity today $\int dm \psi_N(m, t_0) = 1$, while the MF $\psi(m, t_0)$ defined in Eq. (11) in Sec. 2.2.2 is not, since the total mass in PBHs decreases with time due to evaporation. As discussed in Sec. 2.2.2 PBHs with initial mass $m_i \lesssim 1 \times 10^{15}$ g lose a non-negligible fraction of their mass by the present day. Therefore for the evaporation constraints the MF should be evolved to the present day using Eq. (12) before calculating $\psi_N(m, t_0)$ by renormalizing to the present day PBH mass density.

For the evaporation constraints $f_{\text{max}}(m)$ decreases rapidly with decreasing $m$, as the Hawking temperature is inversely proportional to the mass, Eq. (1). Consequently, for sufficiently wide MFs peaked at the lower end of the asteroid-mass range, the contribution to Eq. (13) from $f_{\text{max}}(m)$ at smaller masses than the constraints are publicly available for ($m \lesssim 10^{15}$ g

for the Voyager 1 constraints [22] and $m < 10^{16}$ g for the INTEGRAL/SPI MeV gamma-ray constraints [23]) may be important. At the smallest masses where constraints are publicly available, $f_{\text{max}}(m) \propto m^q$ with $q \approx 2-3$ to a good approximation, and we assume that the power-law form, $f_{\text{max}}(m) \propto m^q$, continues to smaller masses. We have checked that the resulting extended MF constraints do not change significantly (by no more than a factor of a few at peak masses where $f_{\text{PBH}} \sim 1$) if instead $f_{\text{max}}(m)$ becomes constant at small masses.

## 3 Results

In this section we calculate the constraints on the time-evolved lognormal, skew-lognormal, and generalised critical collapse PBH mass functions presented in Sec. 2.2.1, for the constraints reviewed in Sec. 2.1 using the method presented in Sec. 2.3. We do this first for the existing limits in Sec. 3.1, and then for the future prospective limits in Sec. 3.2.

### 3.1 Existing constraints

Fig. 2 shows current constraints on the fraction of DM in PBHs, $f_{\text{PBH}}$, for the fitting functions presented in Sec. 2.2.1 for PBHs arising from a log-normal peak in the power spectrum, Eq. (6), with width $\Delta = 0, 2$ and $5$ ($\Delta = 0$ corresponds to a delta-function peak). These values span the range of values considered by Ref. [16]. As discussed in Sec. 2.1, the constraints we consider are from INTEGRAL/SPI observations of MeV gamma rays [23], Voyager 1 measurements of the local flux of electrons and positrons [22], and the Subaru-HSC microlensing survey [24] as calculated in Ref. [25].

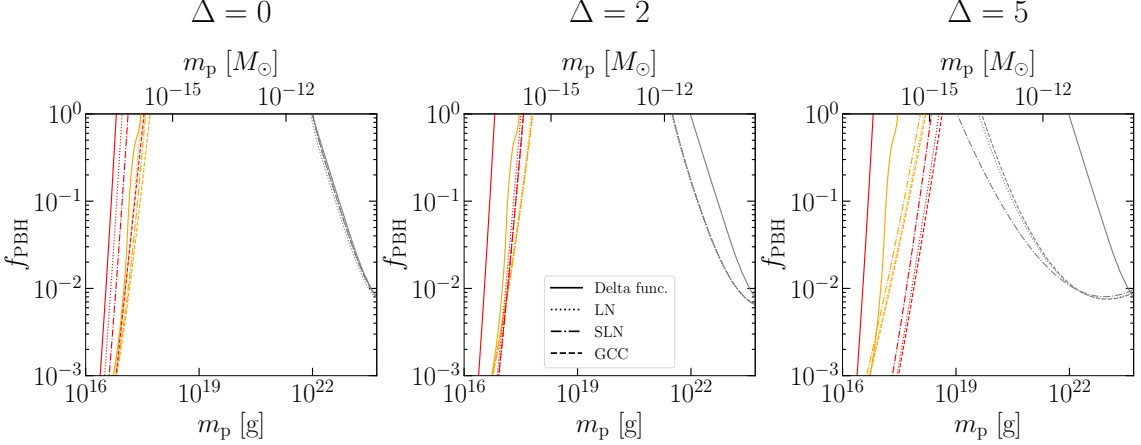

Figure 2: Current constraints on the fraction of dark matter in PBHs, $f_{\text{PBH}}$, as a function of the mass at which the PBH MF peaks, $m_{\text{p}}$, for PBHs formed from a lognormal peak in the primordial power spectrum, Eq. (6), with width $\Delta = 0, 2$ and $5$ (from left to right). Constraints for the lognormal (LN), skew-lognormal (SLN) and generalised critical collapse (GCC) MFs are shown with dotted, dot-dashed, and dashed lines respectively, while the original constraints, calculated assuming a delta-function MF are shown with solid lines. The constraints shown are from Voyager 1 measurements of the local flux of electrons and positrons [22] (red), INTEGRAL/SPI observations of MeV gamma rays [23] (orange), and the Subaru-HSC microlensing survey [24] as calculated in Ref. [25] (grey). In the $\Delta = 2$ case, the constraints for the skew-lognormal and generalised critical collapse MFs are indistinguishable.

As previously seen in e.g. Refs. [6,13], compared to the delta-function MF constraints, the tightest extended MF constraint is weakened, while $f_{\mathrm{PBH}} = 1$ is excluded over a wider range of peak masses $m_{\mathrm{p}}$. As anticipated in Ref. [16], the constraints depend on the shape of the low and high mass tails of the MF. Nevertheless, even for the widest power spectrum considered, $\Delta = 5$, there remains a range of peak masses for which $f_{\mathrm{PBH}} = 1$ is allowed for all three extended mass functions. Our evaporation constraints for extended MFs appear closer to the delta-function MF constraints than previously found for the lognormal MF (see e.g. Fig. 20 of Ref. [6]). As we discuss in App. A, this is largely an artefact of the lognormal MF constraints previously being plotted in terms of the parameter $m_{\mathrm{c}}$ which appears in the definition of the lognormal MF (see Eq. (5)) rather than the peak mass, $m_{\mathrm{p}}$.

For small $\Delta$ the MFs calculated in Ref. [16] are skewed towards low masses and the best-fit lognormal underestimates the low-mass tail and overestimates the high-mass tail. At a given $m_{\mathrm{p}}$, the evaporation and microlensing constraints for a lognormal MF are therefore less and slightly more stringent, respectively, than those for the better fitting skew-lognormal and generalised critical collapse MFs. For $\Delta = 0$, the Voyager 1 constraint for the generalised critical collapse MF (the best-performing function for $\Delta \lesssim 0.5$ [16]) at a given $m_{\mathrm{p}}$ is more stringent than the constraints for the lognormal and skew-lognormal MFs by an order of magnitude or more. This is because the power-law tail of the generalised critical collapse MF at low masses is much larger than the low-mass tails of the lognormal and skew-lognormal MFs, and the constraint from Voyager 1 is especially tight at low $m$. The INTEGRAL/SPI MeV gamma-ray constraints for the different extended MFs are more similar, as this constraint is relatively weak for the range of $m$ where the differences between the MFs are large. Since the microlensing constraints for each MF agree closely, and the extended MF constraints from INTEGRAL/SPI observations of MeV gamma rays are more stringent than those from Voyager 1, the range of $m_{\mathrm{p}}$ where $f_{\mathrm{PBH}} = 1$ is allowed is fairly similar for each MF. For $\Delta = 0$, for the best fitting generalised critical collapse MF, $f_{\mathrm{PBH}} = 1$ is allowed for $5 \times 10^{17}\,\mathrm{g} \lesssim m_{\mathrm{p}} \lesssim 1 \times 10^{22}\,\mathrm{g}$, a slightly narrower mass range than for the lognormal MF.

For $\Delta = 2$, the MF calculated in Ref. [16] is close to symmetric, and all three MFs provide a very good fit [16]. Therefore the constraints for the extended MFs are very similar, with $f_{\mathrm{PBH}} = 1$ being allowed for $6 \times 10^{17}\,\mathrm{g} \lesssim m_{\mathrm{p}} \lesssim 3 \times 10^{21}\,\mathrm{g}$. For $\Delta = 5$, the MF calculated in Ref. [16] is skewed towards large masses and the skew-lognormal MF provides a significantly better fit than the lognormal and generalised critical collapse MFs. For the skew-lognormal MF $f_{\mathrm{PBH}} = 1$ is allowed for $2 \times 10^{18}\,\mathrm{g} \lesssim m_{\mathrm{p}} \lesssim 1 \times 10^{19}\,\mathrm{g}$. The lognormal and generalised critical collapse MFs over (under) estimate the MF at low (high) masses, resulting in overly stringent evaporation (overly weak microlensing) constraints. The range of $m_{\mathrm{p}}$ where $f_{\mathrm{PBH}} = 1$ is allowed is therefore wider and shifted to larger $m_{\mathrm{p}}$ (compared to the better-fitting skew-lognormal MF). For $\Delta = 5$ the strongest evaporation constraints come from Voyager 1, even though for a delta-function MF the INTEGRAL/SPI MeV gamma-ray constraint excludes $f_{\mathrm{PBH}} = 1$ at larger masses than the Voyager 1 constraint (see Fig. 1). This is because for $\Delta = 5$ the MF is sufficiently wide that it is non-negligible in the mass range $m \lesssim 10^{16}$ g where the Voyager 1 constraint is more stringent than the INTEGRAL/SPI MeV gamma-ray constraint, and for the Voyager 1 constraint the integral in Eq. (13) is dominated by this mass range. This highlights that for a broad MF the tightest constraint (i.e. the constraint that rules out $f_{\mathrm{PBH}} = 1$ at the largest peak mass) might not be the constraint which is tightest for a delta-function MF.

For $\Delta \lesssim 2$, the difference between the evaporation constraints calculated using the time-evolved MF $\psi_{\mathrm{N}}(m, t_0)$ and the initial MF $\psi(m_{\mathrm{i}}, t_{\mathrm{i}})$ is no more than 10%, for $m_{\mathrm{p}} \gtrsim 10^{17}$ g. For $\Delta = 5$, the constraints obtained using $\psi_{\mathrm{N}}(m, t_0)$ and $\psi(m_{\mathrm{i}}, t_{\mathrm{i}})$ differ by no more than a factor of two at $m_{\mathrm{p}} = 10^{17}\,\mathrm{g}$ and less than $\approx 20\%$ at peak masses where $f_{\mathrm{PBH}} \sim 1$. For broader mass functions (or for MFs peaked at smaller masses [43]) the effect on the constraints on $f_{\mathrm{PBH}}$ would be larger.

## 3.2 Prospective future constraints

Fig. 3 shows prospective future constraints obtained from MeV gamma-ray telescopes [26] and a proposed microlensing survey of white dwarfs in the LMC [27], as discussed in Sec. 2.1. Due to the improved sensitivity compared to existing observations, $f_{\mathrm{PBH}} = 1$ is excluded over a wider peak mass range than for existing constraints. In particular for the broadest peak in the primordial power spectrum, $\Delta = 5$, $f_{\mathrm{PBH}} = 1$ is excluded across the whole asteroid-mass window for all three MFs, and the maximum allowed PBH dark matter fraction is $f_{\mathrm{PBH}} \sim 0.2 - 0.4$. For $\Delta = 5$ the current Voyager 1 constraint [22] rules out $f_{\mathrm{PBH}} = 1$ at larger $m_{\mathrm{p}}$ than the projected future MeV gamma-ray constraint that we consider, even though the largest mass for which $f_{\mathrm{PBH}} = 1$ is excluded for a delta-function MF is smaller for the Voyager 1 $e^{\pm}$ constraint. As for the current MeV gamma-ray constraint, this is because the low-mass tails of the widest MFs are large at $m \lesssim 10^{16}$ g, where the delta-function MF constraint from Voyager 1 [22] is tighter.

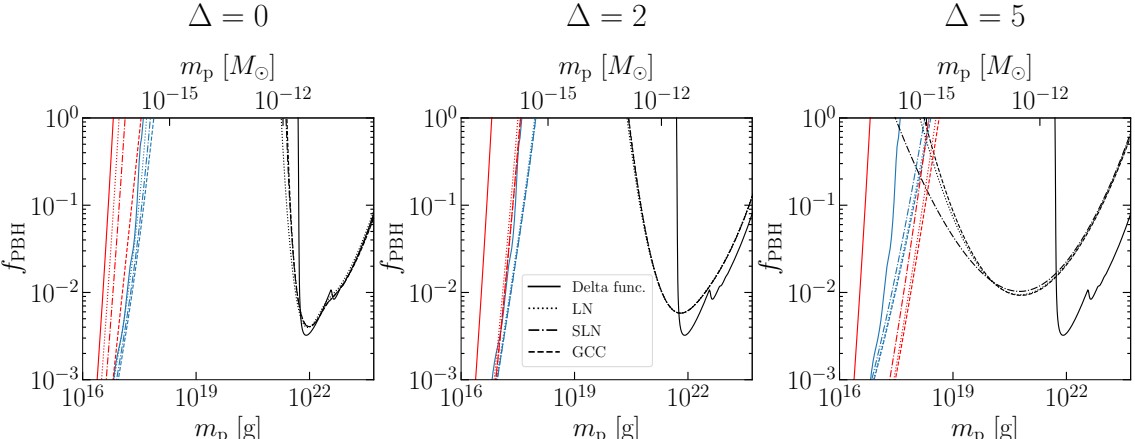

Figure 3: Prospective future constraints on the fraction of dark matter in PBHs, $f_{\mathrm{PBH}}$, as a function of the mass at which the PBH MF peaks, $m_{\mathrm{p}}$, for PBHs formed from a lognormal peak in the primordial power spectrum with width $\Delta = 0, 2$ and $5$ (from left to right). The line styles for the MFs are the same as in Fig. 2. The future constraints shown are from future MeV gamma-ray telescopes (assuming a NFW profile for the Galactic DM halo) [26] (light blue) and stellar microlensing of white dwarfs in the LMC [27] (black). For comparison we also show the current constraints from Voyager 1 measurements of the $e^{\pm}$ flux (red) from Fig. 2.

## 4 Conclusions

If the PBH mass function is a delta-function then PBHs with mass in the asteroid-mass window, $10^{17}$ g $\lesssim m_{\mathrm{p}} \lesssim 10^{22}$ g, can make up all of the DM, i.e. $f_{\mathrm{PBH}} = 1$. However, due to critical collapse, PBHs formed from the collapse of large density perturbations are expected to have an extended MF, even if they form from a narrow peak in the power spectrum. Refs. [6,13] found that the range of masses for which $f_{\mathrm{PBH}} = 1$ is allowed is much smaller for the commonly used lognormal MF than for a delta-function MF. We have explored how constraints on $f_{\mathrm{PBH}}$ in the asteroid-mass window depend on the shape of the PBH MF. In addition to a lognormal MF, we use the skew-lognormal and generalised critical collapse MFs, which Gow et al. [16] found provided a better fit to the MFs they calculated than the lognormal.

We find, using the constraints from Voyager 1 measurements of the local $e^{\pm}$ flux [22], INTEGRAL/SPI observations of MeV gamma rays [23], and microlensing constraints from Subaru-HSC [24, 25], that the asteroid-mass window is typically narrower (i.e. $f_{\text{PBH}} = 1$ is allowed for a smaller range of peak masses, $m_{\text{p}}$) for the better fitting MFs than for the lognormal MF. Nevertheless, for the widest primordial power spectrum considered by Gow et al. [16], there is still a range of $m_{\text{p}}$ values ($2 \times 10^{18}$ g $\lesssim m_{\text{p}} \lesssim 1 \times 10^{19}$ g) where $f_{\text{PBH}} = 1$ is allowed for the skew-lognormal mass function, which is the best-fitting mass function in this case.

The constraint that excludes $f_{\text{PBH}} = 1$ over the widest range of PBH masses for a delta-function MF does not always exclude $f_{\text{PBH}} = 1$ for the widest range of peak masses for extended mass functions. For instance the largest mass for which $f_{\text{PBH}} = 1$ is excluded for a delta-function MF is smaller for the Voyager 1 $e^{\pm}$ constraint than for the MeV gamma-ray constraints (see Fig. 1). However for the widest MFs we consider, the Voyager 1 constraint rules out $f_{\text{PBH}} = 1$ at larger $m_{\text{p}}$ than the current INTEGRAL/SPI MeV gamma-ray constraint and also the projected future MeV gamma-ray constraint that we consider. This shows that tighter constraints on PBHs with $m \lesssim 10^{16}$ g would be beneficial for constraining PBHs with broad MFs.

Future gamma-ray observations will improve limits on the abundance of PBHs with masses $m \lesssim 5 \times 10^{17}$ g, while a proposed LMC white dwarf microlensing survey could provide tighter constraints for $5 \times 10^{21}$ g $\lesssim m \lesssim 2 \times 10^{23}$ g. Together, these constraints could potentially exclude asteroid-mass PBHs with a broad MF making up all of the DM. However the evaporation and microlensing constraints are sensitive to the shape of the low and high mass tails of the MF respectively. An accurate calculation of the shape of the tails of the MF will therefore be essential in future for assessing whether evaporation and microlensing constraints allow asteroid-mass PBHs to make up all of the DM. This also demonstrates the importance of developing new observational probes of PBHs with mass $10^{18}$ g $\lesssim m \lesssim 10^{22}$ g, such as femtolensing of gamma-ray bursts (GRBs) [50, 51], GRB lensing parallax [52, 53], microlensing of X-ray pulsars [54], their effects on stars, e.g. Refs. [35, 55–57], or on the orbits of planets [58] and satellites [59].

# Acknowledgements

We are grateful to Jérémy Auffinger, Andrew Gow and Sunao Sugiyama for useful discussions, and to Andrew Gow for providing the best-fit parameter values for the lognormal mass function.

**Funding information** MG is supported by a United Kingdom Science and Technologies Facilities Council (STFC) studentship. AMG is supported by STFC grant ST/P000703/1.

**Data availability** No new data were created during this study.

# A Comparison with previous constraints for lognormal mass function

Carr and collaborators [6, 13] have previously calculated constraints on $f_{\text{PBH}}$ for a lognormal (LN) MF with width $\sigma = 2$ (see e.g. Fig. 20 of Ref. [6]). Their constraints differ significantly more from the delta-function MF constraints than is the case for the widest lognormal MF we consider, which has $\sigma = 1.8$ [16]. In this appendix we outline the reasons for this apparent difference.

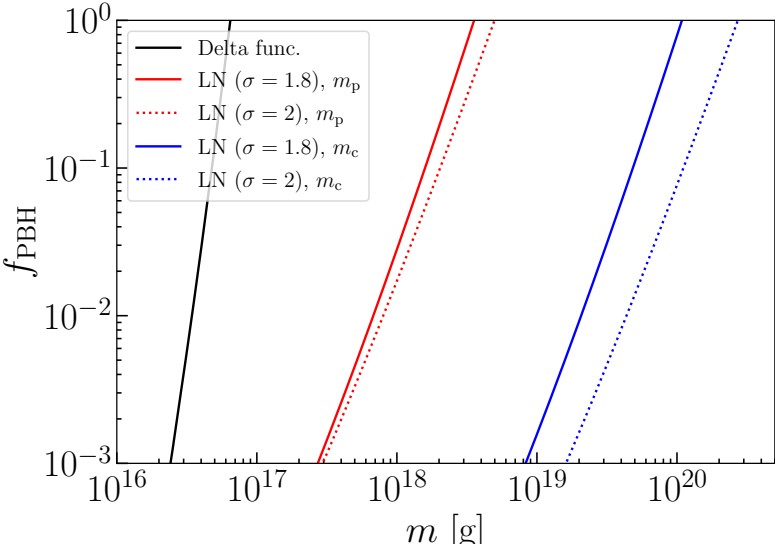

Figure 4: The constraints on the fraction of dark matter in PBHs, $f_{PBH}$, from Voyager 1 measurements of the local flux of electrons and positrons [22] for a delta-function MF and a lognormal (LN) MF. The constraint for a delta-function MF is shown with a black solid line as a function of the PBH mass $m$. The constraint for the lognormal MF, Eq. (5), is shown as a function of both $m_p$ (red) and $m_c$ (blue) for $\sigma = 1.8$ and 2 (solid and dotted lines respectively).

Fig. 4 shows the most stringent constraints on $f_{PBH}$, from Voyager 1 [22], for a lognormal MF with $\sigma = 1.8$ and $\sigma = 2$ plotted as a function of both $m_c$ and $m_p$. The main reason for the apparent difference between our results in Fig. 2 and those in Fig. 20 of Ref. [6] is that the constraints appear significantly different when plotted in terms of the peak mass, $m_p$, than when plotted in terms of the parameter $m_c$, which appears in the definition of the lognormal MF (Eq. (5)). This mass parameter, $m_c$, is the mean of $m_i\psi(m_i, t_i)$ for the lognormal MF and is related to the peak mass by $m_c = m_p \exp(\sigma^2)$, so that for $\sigma \approx 2$, $m_c \approx 50 m_p$. The peak mass better reflects the typical mass of the PBHs, and plotting constraints in terms of the peak mass also allows comparison with other mass functions with a single peak. Furthermore the value for the width of the lognormal used in Refs. [6,13], $\sigma = 2$, is larger than that of the best fit lognormal to the widest power spectrum considered by Gow et al. [16], $\sigma = 1.8$, and this relatively small difference in $\sigma$ leads to a significant shift in the evaporation constraint when it is plotted as a function of $m_c$.

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
