# Peer review of "How open is the asteroid-mass primordial black hole window?"

_SciPost Physics, doi:SciPost Phys. 17, 032 (2024)_

## Round 1 · List of Changes

We are grateful to the referees for their careful reading of the manuscript and helpful comments. Below we respond to their comments and outline the changes we have made. We have highlighted the changes in the manuscript (apart from the expansion of acronyms and new references) in red.

One minor point: the paper is somewhat overladen with acronyms and, while they are all defined, one has to constantly go back to find what they mean.

We now write out lognormal, skew-lognormal, critical collapse and generalised critical collapse in full throughout.

  1. P2. Some researchers would disagree with the claim that solar-mass PBHs are excluded, so they might wish to modify this remark. This is not crucial to the main point of the paper but citing papers with contrary views might be more balanced.

There are modelling uncertainties in the constraints, however we are not aware of any paper that shows that Solar mass PBHs could make up all of the dark matter when all existing constraints are taken into account. In particular, long-duration microlensing surveys (Blaineau et al. (2022) https://inspirehep.net/literature/2040092 and Mroz et al. (2024) https://inspirehep.net/literature/2764863) now place tight constraints on the abundance of compact objects with masses up to 1000 Solar masses. Uncertainties in the distribution of dark matter do not change the excluded mass range by orders of magnitude (see e.g. Alcock et al. (1996) https://ui.adsabs.harvard.edu/abs/1996ApJ...461...84A/abstract, and various subsequent papers). If PBHs formed very compact clusters, so that the cluster as a whole rather than individual PBHs acts as a lens, then the microlensing constraints would be significantly weakened (as in Calcino, Garcia-Bellido and Davis (2018) https://inspirehep.net/literature/1664442). However in this case, as shown in de Luca et al. (2022) (https://inspirehep.net/literature/2131836), other constraints would be tightened such that Solar mass PBHs making up all of the dark matter remains excluded. We have addressed this comment by adding “Under standard assumptions” to the 4th sentence of the introduction, which previously started “PBHs can only account. . . ”

  1. P5. Perhaps the relationship between Refs. 16 and 35 could be clarified. For example, do eqns (7) and (8) both come from Ref. 16? The latter focusses on stellar-mass PBHs but Ref. 35 is earlier and explicitly refers to constraints.

Ref. 35 calculates mass functions numerically, and the results described in the first two sentences of the paragraph starting “Gow et al. [16, 35]” were found in both papers. The subsequent more detailed results (starting from the sentence “As Delta is increased. . . . . . . skewed towards low masses [16]”) are just from Ref. 16. To make this clearer, in the subsequent sentence we have re- placed “Of the fitting functions they consider” with “Of the fitting functions considered in Ref. [16]”.

  1. P6. I think that tau in eqn (10) should have a subscript i to indicate that it’s the lifetime for the PBHs of mass mi.

We have added a subscript ‘i’ to the lifetime tau as suggested.

  1. P7. Although attributed to 2022 papers, the m3 tail effect goes back to the earliest work on PBH evaporation and is crucial (for example) to the analysis of Ref. [39].

We have added citations to Ref. 38 (now Ref. 28) and also Page and Hawking (1976) for the m3 factor in the relationship between the initial and present-day mass functions.

In response to the comments of the 2nd referee we have made the following changes:

  1. The discussion of time variation given in arXiv:2307.06467 should also be mentioned in the text, as should Ref.[40] Mesbeck and Picker.

We’ve added a citation to this paper.

  1. In the section discussing the limit from MeV gamma-rays, I encourage the authors to also mention to Ref.[38], which was the first conservative limit on PBHs by the diffuse MeV-gamma-ray back- ground reported by COMPTEL. This was followed by Ref.[23] Korwar and Profumo and others, who additionally obtained the bound by the galactic gamma-ray limits based on their stronger assumptions although it is debatable whether the more stringent limits obtained based on stronger assumptions, as in Ref.[23], make sense. However, it should be discussed more fairly without omitting the independent bound obtaiened by the diffuse MeV gamma-ray background.

As we tried to explain in the original version of the manuscript (text starting “There are various evaporation constraints, from different particle species and observations, calculated using different assumptions, with different uncertainties...”) we consider two illustrative constraints and aren’t making any statements regarding which constraints are best/most reliable. To address this comment we’ve reworded the aforementioned text and added a citation to Ref. [38] (which is now Ref. [28]) to this paragraph. We’ve also removed “the” from “We calculate the con- straints” in the opening sentence of the final paragraph of Sec. 1 and also from “we have explored how the constraints” in the fourth sentence of the first paragraph of Sec. 4.

Also, in response to comments from readers of the arXiv version, we have • Added a footnote “The proposed AMEGO telescope would be able to exclude fPBH = 1 to somewhat larger m [31,32].” at the end of the final sentence of Sec. 2.1.1. • Added Ref. 57 after “their effects on stars” in the final sentence of Sec. 4.

---

## Editorial Decision

published